# Effects of Flexor Digitorum Longus Muscle Anatomical Structure on the Response to Botulinum Toxin Treatment in Patients with Post-Stroke Claw Foot Deformity

**DOI:** 10.3390/toxins14100666

**Published:** 2022-09-25

**Authors:** Toru Takekawa, Kazushige Kobayashi, Naoki Yamada, Satoshi Takagi, Takatoshi Hara, Tomohide Kitajima, Tomoharu Sato, Hiroshi Sugihara, Kazuo Kinoshita, Masahiro Abo

**Affiliations:** 1Department of Rehabilitation Medicine, The Jikei University School of Medicine, Tokyo 105-8461, Japan; 2Shinagawa Rehabilitation Hospital, Tokyo 141-0001, Japan; 3National Center of Neurology and Psychiatry, Tokyo 187-8551, Japan; 4Department of Cerebrospinal Surgery, Medical Center Narita Hospital, Narita, Chiba 286-0845, Japan; 5Motoyama Rehabilitation Hospital, Kobe 658-0015, Hyogo, Japan; 6Department of Neurology, Kita-Kashiwa Rehabilitation General Hospital, Kashiwa, Chiba 277-0004, Japan; 7The Jikei University Hospital, Tokyo 105-8471, Japan

**Keywords:** cerebrovascular disorders, anatomic variation, hammer toe syndrome, flexor hallucis longus, flexor digitorum longus, electric stimulation, muscle contraction

## Abstract

(1) Background: The purpose of this retrospective case-control study was to determine the relationship between the control of toe movements by flexor hallucis longus (FHL) and flexor digitorum longus (FDL) muscles and the response to treatment with botulinum toxin (BoNT) in post-stroke patients with claw toe. (2) Methods: Subjects with stroke-related leg paralysis/spasticity and claw toes received multiple injections of BoNT (onabotulinumtoxin A) into the FHL or FDL muscles. We investigated the relationship between the mode of transmission of FHL and FDL muscle tension to each toe (MCT) and treatment outcome using the data of 53 patients who received 124 injections with clinically recorded treatment outcome. We also dissected the potential variables that could determine the treatment outcome. (3) Results: The effectiveness of BoNT treatment was significantly altered by FDL-MCT (OR = 0.400, 95% CI = 0.162–0.987, *p* = 0.047). Analysis of the response to the first BoNT injection showed an odds ratio of FDL-MCT of approximately 6.0 times (OR = 0.168, 95% CI = 0.033–0.857, *p* = 0.032). The more tibial the influence of the FDL muscle on each toe, the better the treatment outcome on the claw toe. (4) Conclusions: The anatomic relation between FDL muscle and each toe seems to affect the response to treatment with BoNT in post-stroke patients with claw toes.

## 1. Introduction

We previously advocated for the use of botulinum toxin (BoNT) for the treatment of the claw foot deformity (CFD) by injecting the BoNT into the flexor hallucis longus (FHL) muscle [1,2]. Generally, contraction of the FHL muscle flexes the first toe, while contraction of flexor digitorum longus (FLD) flexes the second to the fifth toes, but we demonstrated in our two recent studies [1,2] that electrically stimulating FHL and FDL muscles results in uneven contraction patterns. However, there is little or no information on the most suitable patients with CFDs for treatment with BoNT.

Local BoNT injection is a common treatment for post-stroke spasticity for lower extremities [3,4,5,6]. The estimated incidence of CFDs in patients hospitalized for rehabilitation after ischemic or hemorrhagic unifocal stroke is 46%, and 83%, respectively, among patients who regain average functional capacities, which start to appear before the end of the third post-stroke month [7]. Thus, the CFD associated with lower limb spasticity is often observed in such patients, and several studies have reported BoNT injection treatment for this condition, where injection targeted the FHL muscle, FDL muscle, flexor digitorum brevis (FDB) muscle, or quadratus plantae (QP) muscles [1,8,9,10,11,12].

The present multicenter, retrospective, case-control study is an extension to our previous studies [1,2], and was designed to identify the effects of FHL and FDL muscle control on the toes. In particular, we determined the relationship between the control of these two sets of muscles and the response to treatment with local injection of BoNT in post-stroke patients with CFDs.

## 2. Results

### 2.1. Overview of All Injections Performed

The study subjects were 58 patients (age, 61.4 ± 10.3 years, males = 46, females = 12). Among all cases, 38 patients suffered from intracerebral hemorrhage (66%), and the rest was cerebral infarction (*n* = 20, 38%). Thirty-eight patients suffered right hemiplegia and 20 from left hemiplegia.

A total of 146 BoNT injections into FHL or FDL were counted during the study period (range: 1–10, median: 2, interquartile range IQR: 1–3) (Figure 1). The age at the time of first injection was 63.4 ± 9.9 years and the latency between stroke onset and first injection was 7.5 ± 4.8 years. The total numbers of BoNT injection into the FHL, FDL, FDB, and QP were 133, 114, 38, and five times, respectively, with a mean total dose of BoNT of 42.9 ± 22.3, 35.3 ± 16.8, 29.9 ± 12.4 and 47.0 ± 2.7 units, respectively.

### 2.2. Classification According to the Induced Toe Movement

#### 2.2.1. Classification about FHL Muscle

Of the 58 patients, 53 received at least one electrical stimulation of the FHL muscle (Figure 1). Classification of the FHL muscle according to the induced toe movement is summarized in Table 1. Type I, in which contraction of this muscle resulted in movement of the big toe only was noted in five (9.4%) patients. Type II, where the stimulation of the FHL muscle led to the movement of the big and the second toes, was noted in 18 (34.0%) patients. However, the majority of the patients (*n* = 30, 57%) were classified as type III to Type V, in whom toe flexion was noted in the third, fourth, and fifth toes.

#### 2.2.2. Classification about FDL Muscle

Of the 58 patients, 53 underwent at least one single session of electrical stimulation of the FDL muscle (Figure 1). Classification of the FDL muscle according to the induced toe movement is summarized in Table 2. The response to such stimulation included movements of toes two to five in 36 patients (68%, Type II). Interestingly, 16 patients showed no movement in the second toe (i.e., Type III to Type V), whereas one patient showed movement of all five toes in response to FDL stimulation (Type I, 1.9%).

### 2.3. The Degree of “Variability” in FHL and FDL Control of Each Toe

Next, we examined the response to FHL muscle stimulation in 30 patients who underwent repeated electrical stimulation (two–10 sessions, median: three) (Figure 1). The mean within-subject standard deviation of the number (hereinafter abbreviated as SD-NSM; the SD of the number of strong toe movements) of the most peroneal toe was 0.53 ± 0.47 (maximum 1.4, minimum 0). On the other hand, 29 subjects underwent repeated FDL electrical stimulation (two to six sessions, median three) (Figure 1). The mean within-subject SD of the number (SD-NSM) of the most tibial toe was 0.52 ± 0.35 (maximum 1.3, minimum 0).

### 2.4. Changes in Subjective Symptoms after BoNT Injection

Changes in subjective symptoms after BoNT injection were reported 83 times in the medical records of patients with cerebral hemorrhage and 41 times by patients with cerebral infarction, after 124 BoNT injections in 53 patients (42 men and 11 women) (Figure 1). The CFD-related signs and symptoms showed improvement in 86 of 124 patients (69%), confirmed by both clinical examination and patients’ self-reports. However, no improvement was observed in the remaining 38 (31%) patients.

### 2.5. The Effects of MCT or NSM on Therapeutic Effects of BoNT Injection

#### 2.5.1. About Therapeutic Effects of All BoNT Injection (n = 124)

Table 3 shows the effects of the mode of control over the toes (hereafter abbreviated as MCT) or the number of strong toe movements (hereafter abbreviated as NSM) on therapeutic effects of all 124 sessions of BoNT injection in 53 patients. There was no significant change in MCT (*p* = 0.653) or NSM (*p* = 0.279) of the FHL muscle. On the other hand, analysis of the FDL muscle showed that the treatment effect significantly changed by MCT (OR = 0.400, CI = 0.162–0.987, *p* = 0.047), but not by NSM (*p* = 0.130). For both muscles, the treatment effect was significantly associated with causal disease (*p* = 0.000). The odds ratio for a 1-point smaller MCT of FDL was approximately 1/0.4 = 2.5 times. The more tibial the influence of the FDL muscle on each toe, the greater the treatment effect on the CFD, and the therapeutic effect on the CFD.

#### 2.5.2. About Therapeutic Effects of the First BoNT Injection (n = 53)

The effects of MCT or NSM of FHL and FDL on the therapeutic effects of the first of the BoNT injections in 53 patients (cerebral hemorrhage: 34, cerebral infarction: 19) are shown in Table 4 (after dividing the patients into those who benefited versus those who did not benefit from the treatment). The MCT of the FHL muscle was not significantly different between the responders (*n* = 31) and non-responders (*n* = 15, *p* = 0.936, Table 4). An analysis of the NSM of the FHL muscle could not be performed due to multicollinearity. In contrast, the MCT of the FDL muscle was significantly different between the responders (*n* = 32) versus the non-responders (*n* = 15) (OR = 0.168, CI = 0.033–0.857, *p* = 0.032, Table 4). The odds ratio for 1-point reduction in the MCT of FDL was approximately 1/0.17 = 6.0 times. Patients who responded to the treatment showed significant muscle contraction and toe movement on the tibial side during electrical stimulation of the FDL muscle. However, there was no significant difference (*p* = 0.090) between the NSM of the FDL muscle in the responders (2.63 ± 0.61, *n* = 32) and non-responders (3.07 ± 0.83, *n* = 14).

Why is the number of subjects in this analysis different from the total number of subjects (*n* = 53)? The reason is that the muscle NSM was not determined when BoNT injection was not performed to the muscle in the first treatment. Furthermore, the MCT of the muscle was not determined when BoNT injection was never done to the muscle in the series of treatments.

### 2.6. Correlation between the NSM of the FHL and the NSM of the FDL

Analysis of the subjects of the first BoNT injection (*n* = 53) showed that 41 had simultaneous injection of BoNT into the FHL and FDL muscles (Figure 1). In these 41 patients, the NSM of the FHL muscle correlated significantly with the NSM of the FDL muscle (r = 0.395, *p* = 0. 011). The more tension the FHL muscle exerted on the peroneal side of the toe, the less tension the FDL muscle exerted on the tibial side of the toe. The regression equation was as follows.
(NSM of FDL muscle) = 1.878 + 0.350 × (NSM of FHL muscle)
(R-squared = 0.156, *p* = 0.011)

## 3. Discussion

We demonstrated in this study that the outcome of BoNT treatment for post-stroke CFD depends on the effect of FDL on the individual toes; the treatment was more effective when contraction of the FDL muscle controlled the movement of more than one toe, such as the second (the tibial side) to fifth toe. BoNT treatment of the CFD was more effective 2.5 to 6.0 times when the action of the FDL muscle extended by one toe on the tibial side. This finding was statistically adjusted for the confounding factor of causative disease.

The FDL muscle is small compared to the space/volume of the lower leg, and for this reason an electrical stimulator, an electromyogram (EMG) measuring device, or an echo device is required to reliably administer the drug to the muscle belly. In recent years, it has become more common to administer drugs in a more non-invasive manner using echo devices. In this case as well, if there is movement in the cross-section of the muscle on the screen when each toe is passively moved one by one in order while observing the FHL and FDL muscles with an echo device, it can be inferred that the muscle is transmitting tension to that toe. Therefore, this is a clinically useful method of administering drugs without using an electrical stimulator. We have performed many botulinum toxin treatments for CFD, and we have experienced only one side effect of excessive extension of the first toe after administering the drug to the FHL muscle, but we have never experienced such a side effect with the FDL muscle.

In the present study, we used two indices to classify the mode of control of the FHL and FDL muscles on each toe. MCT (the mode of control over the toes) is a qualitative variable that provides ordinal scale in the individual patient. In other words, it is a single index that reflects the manner in which the FHL or FDL muscle controls the toe movement. On the other hand, the NSM (the number of strongly moving toes) is a quantitative variable that represents the number of patient’s toes that moved in response to each BoNT injection. The same patient may show variable response each time in multiple drug administrations of BoNT, and thus the same patient can have several different numbers of these indices. It seems that MCT rather than NSM is related to the effectiveness of the treatment. Thus, rather than focusing on inducing more toe movements with electrical stimulation of the FDL muscle during BoNT injection, the patient-specific mode of dominance of the FDL muscle over the individual toes may determine the outcome of BoNT treatment in post-stroke patients with CFD. In this study, we could not find any specific factor that can be controlled by the therapist and that influenced the outcome of treatment. We could not elucidate how to treat patients whose FDL muscles control only the fewer toes to obtain a better therapeutic effect. Future studies need to be more objective and detailed in their assessment of the effects of BoNT treatment in this population of patients. Furthermore, regarding long-term treatment strategies, we may suggest the administration dose over time of the drug.

Although they had no effect on the conclusions of this study, analyzing the relationship of NSM of the FHL muscle and the causative disease and treatment outcome suggested its multicollinearity, probably due to the association between the underlying condition and the treatment outcome in patients with documented NSM of the FHL. We presume the multicollinearity is related to the effects of the less-than-ideal methodology used to determine the treatment outcome and the relatively small number of patients.

Toe movement varies each time with multiple electrical stimulations of the FHL and FDL muscles. In order to investigate such variability in the present study, we used the term “variability”, which reflected the standard deviation of the number of toes that moved following muscle contraction within the same subject (= SD-NSM). Therefore, the standard for “variability” is 0 if movement occurred in the same toe following each muscle contraction, and 0.7 if movement included adjacent toes. During electrical stimulation of the FHL muscle, the above variability index (SD-NSM) in the same subject was 0.53, while that following electrical stimulation of the FDL muscle SD-NSM was 0.52. These results suggest that the range of “variation” for both the FHL and FDL muscles is generally the adjacent toe or less. The within-subject variability was modest.

Since the FHL muscle branches out to the FDL muscle as well as to the big toe, we believe that treatment of the second and third CFDs with BoNT should involve the injection of BoNT into the FHL muscle as well as the FDL muscle [1,2]. In addition to it, we now recognize that the more toes that move with FDL muscle stimulation, the better the outcome would be with the BoNT treatment, giving us a better prediction of the treatment outcome, as well as enabling us to avoid unnecessary drug injections to the patients.

In this study, we categorized the dominant type of the FHL and FDL muscles for each toe by analyzing the results of multiple sessions in patients who underwent repeated electrical stimulations. The results showed fewer type IIs and more type III to type V in FHL compared to the findings of previous studies [1,2]. The insertion of the FHL muscle into each toe naturally affects the big toe, but it also affects the second and the more peroneal toes.

The interconnections between the FHL and FDL tendons are well known. The FHL tendon separates into two parts, and one joins with the FDL tendons. This tendinous crossover has been examined previously and is known as “the Knot of Henry” [13] or “Junctura Tendinum” [14]. The Knot of Henry is can be found 1.8 cm below the navicular tuberosity and 5.9 cm distal to the medial malleolus [15]. Although the above ratios varied in individual studies, the majority reported a pattern of tendon binding from FHL to FDL. In our previous studies [1,2], the dissection of six cadavers (twelve legs) showed that the FHL tendon bifurcated and joins the FDL in all of them, while there was no tendon coupling from the FDL to the FHL (Figure 2a–d). In one previous study examining the cadavers’ Knots of Henry in 16 limbs [13], contraction of the FHL in 11 limbs resulted in not just the big toe motion but the second toes as well as the toes on peroneal side, while contraction of the FDL induced no flexion of the big toe. They also mentioned that the transmission of the tension was unidirectional and was always from the FHL to FDL. In a similar study involving 24 limbs of 24 cadavers [16], 10 limbs showed that the FHL tendon branched into the FDL (42%). In another study, the dissection of the Knot of Henry in 20 limbs of 10 cadavers [15] showed that the tendon overlap was unidirectional, from the FHL to the FDL in 15 limbs (75%). Another study [17] reported a unidirectional tendon from FHL to FDL in all 50 cadavers. Furthermore, a recent study [18] found a unidirectional tendon from FHL to FDL in 97% of 100 lower limbs in 55 cadavers. Thus, we could assume that FHL greatly affects the movement of the second and/or the third toes [19].

On the other hand, in not a few cases, the FDL does not move the second toe [1,2]. We speculate that this is due to maldevelopment and follows the pattern of the Knot of Henry described above. In these cases, it likely represents the progression of FHL muscle dominance on movements of the second and third toes, with simultaneous and relative regression of the FDL muscle dominance. In fact, our results showed a significant correlation between the NSM of the FHL muscle and the NSM of the FDL muscle, adding support to our speculation.

The present study has several limitations. First, the assessment of CFD before BoNT injection did not include a detailed evaluation of mAS [20] of the toes or objective confirmation of the appearance of CFD by visual inspection during movement. Evaluation of the treatment outcome was limited to face-to-face interviews of subjective symptoms by the therapist with the patient; only subjective symptoms, such as pain at rest or during transfer or walking, were evaluated. Second, our study was retrospective in nature, and the main goal of each BoNT treatment was not to improve the CFD, and it is possible that sufficient doses of BoNT were not selected for treatment of the CFD. Naturally, each toe was not targeted separately during injection either. Third, the study subjects included those with BoNT injections into the flexor digitorum brevis muscle but not those who got orthosis therapy after BoNT treatment for CFD. Fourth, while the treatment effects on the CFD was examined, we did not take into account the possible contribution of tenodesis on the ankle position to the CFD. Many patients were in position with plantar flexion and ankle inversion, for whom further BoNT was injected simultaneously into the triceps surae and tibialis posterior muscles. The fifth limitation is the varying timing of assessment. The assessment time of the treatment outcome varied from two to six weeks, and thus we may have missed clinical changes in some patients. These points are issues that need to be considered in future studies.

## 4. Conclusions

Treatment of the CFD with BoNT was effective in about 68% of cases. The way the FDL muscles control each toe differs from patient to patient, and those with the FDL muscle controlling more toes are more likely to benefit from treatment than those with fewer toes control. The study results suggest that we could predict the effectiveness of BoNT treatment based on how many toes move with FDL stimulation. The treatment outcome was probably affected, at least in part, by the dominance of FDL muscle control on the movement of each toe. The FHL muscle often affects the second toe and below, while the FDL muscle does not always control the second to fifth toes, and such correlation between the two was evident in this study. The movement of each toe during electrical stimulation of the FHL and FDL muscles varied with each electrical stimulation, but the range of variation was generally within the adjacent toe or less.

## 5. Materials and Methods

The protocol of this study was approved by the Ethics Review Committee of the Jikei University School of Medicine for Biomedical research [#26-377(7883)]. Each patient had signed the consent form for BoNT administration before the injection. The study was carried out in compliance with the Helsinki Declaration.

The inclusion criteria were as follows: (1) more than 12 months of history of stroke-related lower limb paralysis its spasticity graded 1 or more on the modified Ashworth scale (mAS) [20]; (2) Previous diagnosis of “CFD due to spasticity” at the Outpatient Department; (3) Age > 20 years; (4) time between the onset of stroke and current treatment of ≥three months; (5) No contraindication to BoNT administration [21,22]; and (6) Previous BoNT injection to the FHL or FDL muscles, along with the electrical stimulation of CFD, between 1 August 2013 and 28 February 2021. The data of patients who opted out of the study were excluded from analysis.

The medical history of each subject was collected from the following six facilities in Japan: The Jikei University Kashiwa Hospital (Kashiwa, Chiba, Japan), The Jikei University Daisan Hospital (Komae, Tokyo, Japan), Tokyo Teishin Hospital (Tokyo, Japan), Medical Center Narita Hospital (Narita, Chiba, Japan), Kita-Kashiwa Rehabilitation General Hospital (Kashiwa, Chiba, Japan), and the Motoyama Rehabilitation Hospital (Kobe, Hyogo, Japan).

The analyzed patients were those for whom we performed BoNT injection into the spastic muscles for the upper/lower extremity of the paralytic side (Figure 1). Before administration, we diluted BoNT with saline down to 1.25–2.5 units/0.1 mL, then injected with neuromuscular electoral stimulation as guidance. We used 25-gauge sterile pole anesthesia needles (Top Co., Tokyo, Japan) for FHL and FDL muscle injections. The number of BoNT units injected into the FHL, FDL, flexor digitorum brevis (FDB), and quadratus plantae (QP) muscles was estimated from the medical records.

We applied the electoral stimulation using a New Tracer NT-11 (Top Co., Tokyo, Japan). The sites and directions of the needle insertion are shown as in Figure 3. For each patient, we confirmed that the stimulation needle was in the right place (i.e., FHL or FDL muscles) by virtually recognizing the muscle contraction/toe movement during the stimulation. A precaution was taken not to inject the BoNT to the tibial nerve, located close to the FHL muscle. Because it would create a painful muscle contraction, we judged that the tibial nerve was stimulated when/if the entire foot contracted, unlike when the FHL or FDL muscle was focally stimulated. Two physicians or a physician and a nurse observed and recorded the effects of FHL/FDL muscle contraction on toe movements.

In the next step, we evaluated the number of toes controlled by the contraction of FHL and FDL based on differences in the response to electrical stimulation. We retrospectively sorted the muscle contractions by their strength: + (marked muscle contraction), ± (weak contraction), and − (no contraction), as described previously [1,2]. Furthermore, we also analyzed the number of toes that moved with the electrical stimulation of the FHL muscle and classified them. Type I represented movement of the first toe only upon FHL stimulation, type II was for the first and second toes, type III was for the first to third toes, and type IV was for the first to fourth toes. Furthermore, we used “>” if the contraction was weak on the peroneal side of the stimulated toes, and put no sign if it was strong. Similarly, for the FDL muscle, we classified them into type I as those with movement of the first to fifth toes, type II of the second to fifth toes, type III of the third to fifth toes, and type IV of the fourth and fifth toes only. Similar to the FHL muscle, we put no sign if a strong contraction on the most tibial side of the toes and the fifth toe was recorded. We put “<” for weak contraction in the most tibial side of the toes, and weak contraction of the fifth toe, i.e., the most peroneal side, was marked as “>” [2].

We used the median value in patients who underwent repeated tests described above on the control of toe movement. To calculate the median value, the numbers were placed in the following order: I, >II, II, >III, III, >IV, IV, and >V for FHL; and I, <I, >II, II, <>II, <II, III, <>III, <III, IV, <>IV, <IV, and V for FDL. When the median value could not be determined, the most frequent value was used in evaluation of the state of control for the patient in question, and when even that could not be determined, we used the number of the first evaluation to determine the state of control. Based on these procedures, we determined the state of control (qualitative variables and ordinal scales) for each toe of the FHL and FDL muscles for all patients (MCT: the mode of control over the toes). However, if one of the two muscles was never stimulated, it was not possible to determine the state of control of that muscle over the toe in that patient.

In the next step, we assessed the degree of “variability” in FHL and FDL control of each toe using the data of patients who underwent several toe muscle stimulation tests. In this procedure, weak movements of the toes were ignored, and the number of strong toe movements was used as a quantitative variable (NSM: number of strong toe movements). The standard deviation of the NSM (SD-NSM) was used to assess the “variability” within the same subject. For FHL and FDL muscles, we calculated the mean value and standard deviation of SD-NSM using the data of all subjects who underwent repeated electrical stimulation tests.

The medical records of the participants were searched for improvements or worsening of clinical signs and symptoms of CFD at 14–42 days after BoNT injection. The medical records were written based on the attending physicians’ clinical assessment as well as on the direct interview with the patients/relatives. The assessor was blinded to the results of FHL and FDL electrical stimulation and the underlying clinical condition of the patient. Specifically, we collected data on post-treatment subjective symptoms (124 BoNT injections in 53 patients) from the 146 data points (Figure 1). Next, in the 124 BoNT injections (with treatment outcome documented in the medical records), we compared and examined the relationship between MCT and NSM and the effects of BoNT injections into the FHL and FDL muscles, respectively, including the effects in those patients who received several injections. Furthermore, analysis of the FDL muscle included an adjustment for factors (underlying conditions) that may have an impact on the objective variable in order to remove confounding (Table 3).

We then examined the relationship between MCT and NSM and the efficacy of the first single BoNT injection in 53 patients (Figure 1) after adjustment for factors related to underlying conditions that could affect the objective variable in particular in order to remove confounding (Table 4). Finally, for the same patients, we examined the correlation between the NSM of FHL and FDL muscles by simple regression analysis.

Baseline subject characteristics are presented as frequencies and proportions for categorical data, and summary statistics (number of subjects, mean ± standard deviation, or median with interquartile range) are presented for continuous data. Logistic regression analysis was used to compare two groups of patients with and without subjective efficacy of initial treatment for the MCT and NSM of the two muscles. The treatment outcome was the objective variable (validity: 1, invalid: 0), and the MCT or NSM of the FHL or FDL muscle and the causative disease (cerebral hemorrhage: 1, cerebral infarction: 0) were the explanatory variables. The generalized estimating equations (GEEs) for the logistic regression model were used to test for all repeated BoNT injections in the same patient, including the second and subsequent injections. The presence or absence of treatment effect (validity: 1, invalid: 0) was used as the objective variable. The model was adjusted for the MCT and NSM for FHL or FDL muscle and the underlying condition (cerebral hemorrhage: 1, cerebral infarction: 0). Pearson’s correlation coefficient was used to examine the correlation between the NSM of the FHL muscle and the NSM of the FDL muscle at the time of the first injection. Two-tailed tests were performed and *p* values of <0.05 denoted the presence of statistically significant differences. All analyses were performed using the SPSS statistics software (ver. 26, IBM Japan, Tokyo).

## Figures and Tables

**Figure 1 toxins-14-00666-f001:**
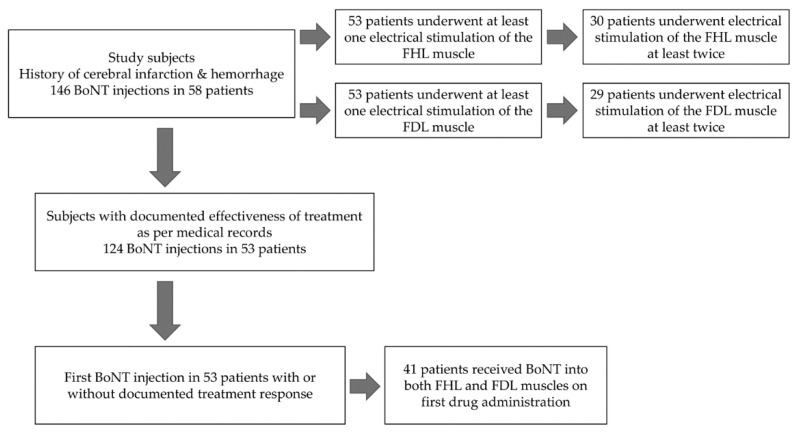
Study subjects.

**Figure 2 toxins-14-00666-f002:**
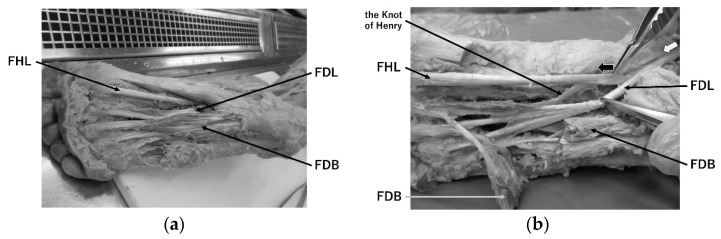
All figures are the right leg of a man who died at the age of 86. His cause of death was stomach cancer. He had a history of benign colorectal polypectomy, cardiac pacemaker implantation, and a right humerus fracture from a car accident. He has not undergone surgery for gastric cancer. The thick black and white arrows in figures (**b**–**d**) indicate the direction of tension transmission in the FHL and FDL muscles, respectively. (**a**) The figure after excision of the aponeurosis plantaris, with the thick-walled FDB muscle in the center. (**b**) The FDL muscle is exposed after excision of the proximal part of the FDB muscle and its distal part is rotated. The FHL muscle branches into two branches immediately after overlapping the FDL muscle, with the main trunk stopping at the big toe. The other, slightly thinner branch, “the Knot of Henry”, joins the FDL and stops on the side of the second toe. (**c**) The FDL muscle is transected near the ankle joint, and this is deflected downward; the area where “the Knot of Henry” joins the FDL is shown. (**d**) The FHL is transected just above its bifurcation into the FDL, and “the Knot of Henry”, where the FHL bifurcates, is more clearly shown.

**Figure 3 toxins-14-00666-f003:**
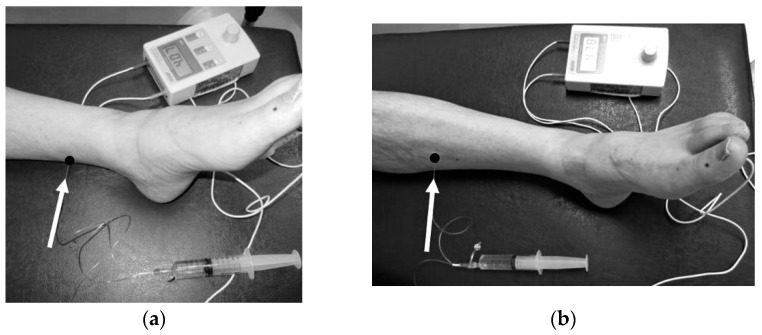
(**a**) Injection site and needle insertion direction into the FHL muscle. White arrow: direction of needle insertion, black circle at end of white arrow: area of needle insertion. (**b**) Injection site and needle insertion direction into the FDL muscle. White arrow: direction of needle insertion, black circle at end of white arrow: area of needle insertion.

**Table 1 toxins-14-00666-t001:** Classification of FHL dominance over each toe (53 patients).

NSM	1st Toe	2nd Toe	3rd Toe	4th Toe	5th Toe	MCT	
1	+	-	-	-	-	I	
							5
1	+	±	-	-	-	>II	
							1
2	+	+	-	-	-	II	
							17
2	+	+	±	-	-	>III	
							2
3	+	+	+	-	-	III	
							19
3	+	+	+	±	-	>IV	
							4
4	+	+	+	+	-	IV	
							3
4	+	+	+	+	±	>V	
							2
5	+	+	+	+	+	V	
							0
						total number	53

The table shows the presence or absence of muscle contraction of each toe evoked by electrical stimulation to FHL muscle. +: muscle contraction, ±: weak muscle contraction, -: no muscle contraction. The mode of control over the toes (MCT) values correspond to the mode of muscle contraction of each toe. For reference, we added the classification of number of strong movements of the toe (NSM) corresponding to the mode of muscle contraction of each toe. Based on the classification of MCT, the NSM corresponding to the same row is not uniquely determined. For example, if the first toe is: +, the second toe is: +, the third toe is: ±, the fourth toe is: ±, and the fifth toe is: -, then the MCT is >IV and the NSM is 2.

**Table 2 toxins-14-00666-t002:** Classification of FDL dominance over each toe (53 patients).

NSM	1st Toe	2nd Toe	3rd Toe	4th Toe	5th Toe	MCT	
1	+	+	+	+	+	I	
							0
2	±	+	+	+	+	<I	
							1
2	-	+	+	+	±	>II	
							2
2	-	+	+	+	+	II	
							16
3	-	±	+	+	±	<>II	
							0
3	-	±	+	+	+	<II	
							18
3	-	-	+	+	+	III	
							12
4	-	-	±	+	±	<>III	
							1
4	-	-	±	+	+	<III	
							0
4	-	-	-	+	+	IV	
							1
5	-	-	-	±	±	<>IV	
							1
5	-	-	-	±	+	<IV	
							0
5	-	-	-	-	+	V	
							1
						total number	53

The presence (+, ±) or absence (-) of reactive muscle contraction in each toe evoked by electrical stimulation to FDL muscle. +: with muscle contraction, ±: with weak muscle contraction, -: without muscle contraction. MCT (the mode of control over the toes) values correspond to the mode of muscle contraction of each toe. For reference, we added the classification of NSM (number of strong movements of the toe) corresponding to the mode of muscle contraction of each toe. Based on the classification of MCT, the NSM corresponding to the same row is not uniquely determined. For example, if the first toe: -, the second toe: ±, the third toe: ±, the fourth toe: +, the fifth toe: +, then the MCT is <II and the NSM is 4.

**Table 3 toxins-14-00666-t003:** Generalized estimation equations were used to compare the presence or absence of treatment effect for a total of 124 BoNT injections in 83 post-hemorrhage patients and 41 post-stroke patients.

	OR	95% CI	*p* Value
MCT of FHL	0.883	0.512–1.522	0.653
NSM of FHL	0.778	0.494–1.226	0.279
MCT of FDL	0.384	0.176–0.837	0.016
NSM of FDL	0.601	0.372–0.970	0.037
MCT of FDL	0.400	0.162–0.987	0.047
disease	5.787	2.369–14.134	0.000
NSM of FDL	0.685	0.419–1.119	0.130
disease	3.306	1.699–6.433	0.000

Results of generalized estimation equations for treatment outcome (with effect: 1, without effect: 0). Effect of background disease (cerebral hemorrhage: 1, cerebral infarction: 0). OR: odds ratio, 95% CI: 95% confidence interval.

**Table 4 toxins-14-00666-t004:** Logistic regression analysis was used to compare the presence or absence of treatment effect for a total of 53 patients treated with BoNT (34 post-cerebral hemorrhage and 19 post-stroke patients).

	OR	95% CI	*p* Value
NSM of FHL	*		
disease			
MCT of FHL	1.031	0.484–2.196	0.936
disease	0.054	0.006–0.473	0.008
NSM of FDL	0.404	0.142–1.151	0.090
disease	5.712	1.056–30.884	0.043
MCT of FDL	0.168	0.033–0.857	0.032
disease	8.732	1.379–55.315	0.021

* Statistical analysis could not be performed due to multicollinearity. Results of logistic regression analysis for treatment outcome (with effect: 1, without effect: 0). Effect of background condition (Disease: cerebral hemorrhage: 1, cerebral infarction: 0). OR: odds ratio, 95% CI: 95% confidence interval.

## Data Availability

Not applicable.

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
