# Peer review of "Effects of Flexor Digitorum Longus Muscle Anatomical Structure on the Response to Botulinum Toxin Treatment in Patients with Post-Stroke Claw Foot Deformity"

_toxins, 2022, doi:10.3390/toxins14100666_

Round 1

Reviewer 1 Report

The paper is detailed and generallywell writtehn. there is a great degree of discussion about the anatomical connectivity and the clinical effects of injectios of the two different muscles. As a topic of academic value the paper is interesting.

My main concern is the clinical relevance.

1. Is there truly a way in targeting the fascicular controls of each individual toe with FDL? This muscle is hard to inject at the best of times

2. Why does it matter to have such precision in the injection? If all the toes are affected by BoNT, does it cause untoward complications such as excessive toe extension? I have never seen this as a side effect.

3. The discussion on FDL and FHL connectivity is interesting. Do the authors suggest a way of injecting that is different due to this? Do they favor injecting one more than the other? Otherwise I dont see value in the detailed discussion of the Knot of Henry.

4. The topic of the etiology of CFD stroke versus hemorrhage appears to be very weak to me. Although there is literature on this, I dont feel that this can really be justified and I would remove it and group them all together anyways.

5. Most imortantly, I think that main weakness of the paper is that there is substantial variablity in patients between injections, after injections and also between patietns. Again regarding the clinical relevance, I am not fully convinced this will change anything in regards to how I inject.

Author Response

Manuscript ID: toxins-1888326

Title: Effects of flexor digitorum longus muscle anatomical structure on the response to botulinum toxin treatment in patients with post-stroke claw foot deformity

Dear 

Thank you for your e-mail regarding the decision on the above manuscript. We were pleased to know of your positive evaluation of our manuscript and its potential acceptance for publication in Toxins, subject to adequate revision and response to the reviewer's comments. 

Based on your instructions, we logged into the Editorial Manager website and submitted the marked up file of the revised manuscript (file name: toxins-1888326-R1m), the clear file of the revised manuscript (file name: toxins-1888326-R1) and the file of the point-by-point response to the comments raised by the reviewers (file name: toxins-1888326-rev) in Microsoft Word format.

Appended to this letter is our detailed point-by-point response to the comments raised by the reviewers. We agreed with all the comments. We revised the text of the discussion section primarily according to the reviewer's suggestions, especially with regard to the clinical relevance. In addition, in order to make the manuscript is easier to read, we revised the English text a little, taking care not to change the main findings and discussion of the results.

We take this opportunity to express our gratitude to the reviewer for the constructive and useful remarks. The comments allowed us to identify areas in our manuscript that needed modification and clarification. We also thank you for allowing us to resubmit a revised copy of the manuscript.

I hope that the revised manuscript is now acceptable for publication in Toxins.   

Sincerely Yours,

Manuscript ID: toxins-1888326

Title: Effects of flexor digitorum longus muscle anatomical structure on the response to botulinum toxin treatment in patients with post-stroke claw foot deformity

Point-by-point response to the comments of Reviewer #1

We thank the reviewer for evaluating our manuscript. The following text describes our response to the comments made by the reviewer. All line numbers mentioned in each response to each comment refer to the small-size numbers that appear on the left margin of the text of the revised manuscript.

Comments and Suggestions for Authors:

The paper is detailed and generallywell writtehn. there is a great degree of discussion about the anatomical connectivity and the clinical effects of injectios of the two different muscles. As a topic of academic value the paper is interesting.

My main concern is the clinical relevance.

We thank the reviewer for the positive evaluation of our manuscript.

The manuscript was revised according to the comments raised by the reviewer. Furthermore, we have reexamined the clinical relevance of the manuscript in accordance with your suggestion, and have added and revised mainly the discussion part of the manuscript.

Comment 1: Is there truly a way in targeting the fascicular controls of each individual toe with FDL? This muscle is hard to inject at the best of times

(Our response to the reviewers' comments) We appreciate the insightful comments. Based on the comment, we will add this point to the limitations of this study as follows. Naturally, each toe was not targeted separately during injection either. Because there is no way to target each toe separately when injecting into the FDL muscle as you pointed out. (Page 7, Line 265-266)

Comment 2: Why does it matter to have such precision in the injection? If all the toes are affected by BoNT, does it cause untoward complications such as excessive toe extension? I have never seen this as a side effect.

(Our response to the reviewers' comments) We thank the reviewer for the comment. At the time this study was conducted, we did not have an echo system, so we used an electrical stimulator to administer the drug. This research can be said to be a by-product of that. Based on the comment, we have added the following statement to the discussion section of the manuscript. The FDL muscle is small compared to the space/volume of the lower leg, and for this reason an electrical stimulator, an electromyogram (EMG) measuring device, or an echo device is required to reliably administer the drug to the muscle belly. We have performed many botulinum toxin treatments for CFD, and we experienced only one side effect of excessive extension of the first toe after administering the drug to the FHL muscle, but we have never experienced such a side effect after administering to the FDL muscle. (Page 6, Line 170-172; Line 178-181)

Comment 3: The discussion on FDL and FHL connectivity is interesting. Do the authors suggest a way of injecting that is different due to this? Do they favor injecting one more than the other? Otherwise I dont see value in the detailed discussion of the Knot of Henry.

(Our response to the reviewers' comments) We appreciate the insightful comments. As we have suggested in our previous reports, we recommend that when targeting CFD in the second or third toe, the drug should be administered to the FHL muscle as well as the FDL muscle, as this may enhance the therapeutic effect. Based on the comment, we have added the following statement to the Discussion and Conclusions section of the manuscript.

Rather, we now recognize that the more toes move with FDL muscle stimulation, the better the outcome would be with the BoNT treatment, giving us a better prediction of the treatment outcome as well as enabling us to avoid unnecessary drug injection to the patients.

The way the FDL muscles control each toe differs from patient to patient, and those with the FDL muscle controls more toes are more likely to benefit from treatment than those with fewer toes control. This study results suggest that we could predict the effectiveness of BoNT treatment based on how many toes move with FDL stimulation.

We have stated these points more clearly in the text. (Page 7, Line 221-224; Page 8, Line 290-294)

Comment 4: The topic of the etiology of CFD stroke versus hemorrhage appears to be very weak to me. Although there is literature on this, I dont feel that this can really be justified and I would remove it and group them all together anyways.

(Our response to the reviewers' comments) Thank you for your comment. Regarding the etiology of cerebral infarction and hemorrhage, we have deleted all the descriptions about it from the text, as you pointed out, since they are not directly relevant to the conclusion. (Page 4, Line 121; Page 6, Line 168)

Comment 5: Most imortantly, I think that main weakness of the paper is that there is substantial variablity in patients between injections, after injections and also between patietns. Again regarding the clinical relevance, I am not fully convinced this will change anything in regards to how I inject.

(Our response to the reviewers' comments) Thank you for your comment, and we agree to what you have pointed out. Based on the comment, we have added the following to the discussion and conclusions section.

Regarding clinical relevance, our main points in this manuscript are that the way the FDL muscles control each toe differs from patient to patient, and that patients in whom the FDL muscle controls more toes are more likely to benefit from treatment than those with fewer toes control. Fortunately, the within-subject variability was not as great as in the adjacent toes, as we mentioned in the text (Results: 2.3). Unfortunately, this study did not elucidate how to treat patients whose FDL muscles control only the fewer toes to obtain a better therapeutic effect. We added to this point in the text. (Page 6, Line 195-197; Page 7, Line 216; Page 8, Line 290-293)

Reviewer 2 Report

The manuscript "Effects of flexor digitorum longus muscle anatomical structure 2 on the response to botulinum toxin treatment in patients with 3 post-stroke claw foot deformity" is well represented and the figures and diagrams expose their concept clearly. Botulinum toxin is used for various pathologies, especially on spasticity. I agree with the limitations set out by the authors especially for side effects and there is no long-term evidence. Regarding this, the authors can suggest the administration dose over time of the drug.

Author Response

Manuscript ID: toxins-1888326

Title: Effects of flexor digitorum longus muscle anatomical structure on the response to botulinum toxin treatment in patients with post-stroke claw foot deformity

Dear 

Thank you for your e-mail regarding the decision on the above manuscript. We were pleased to know of your positive evaluation of our manuscript and its potential acceptance for publication in Toxins, subject to adequate revision and response to the reviewer's comments. 

Based on your instructions, we logged into the Editorial Manager website and submitted the marked up file of the revised manuscript (file name: toxins-1888326-R1m), the clear file of the revised manuscript (file name: toxins-1888326-R1) and the file of the point-by-point response to the comments raised by the reviewers (file name: toxins-1888326-rev) in Microsoft Word format.

Appended to this letter is our detailed point-by-point response to the comments raised by the reviewers. We agreed with all the comments. We revised the text of the discussion section primarily according to the reviewer's suggestions, especially with regard to the clinical relevance. In addition, in order to make the manuscript is easier to read, we revised the English text a little, taking care not to change the main findings and discussion of the results.

We take this opportunity to express our gratitude to the reviewer for the constructive and useful remarks. The comments allowed us to identify areas in our manuscript that needed modification and clarification. We also thank you for allowing us to resubmit a revised copy of the manuscript.

I hope that the revised manuscript is now acceptable for publication in Toxins.   

Sincerely Yours,

Manuscript ID: toxins-1888326

Title: Effects of flexor digitorum longus muscle anatomical structure on the response to botulinum toxin treatment in patients with post-stroke claw foot deformity

Point-by-point response to the comments of Reviewer #2

We thank the reviewer for evaluating our manuscript. The following text describes our response to the comments made by the reviewer. All line numbers mentioned in each response to each comment refer to the small-size numbers that appear on the left margin of the text of the revised manuscript.

Comments to the authors:

The manuscript "Effects of flexor digitorum longus muscle anatomical structure 2 on the response to botulinum toxin treatment in patients with 3 post-stroke claw foot deformity" is well represented and the figures and diagrams expose their concept clearly. Botulinum toxin is used for various pathologies, especially on spasticity. I agree with the limitations set out by the authors especially for side effects and there is no long-term evidence. Regarding this, the authors can suggest the administration dose over time of the drug.

We thank the reviewer for the positive evaluation of our manuscript.

The manuscript was revised according to the comments raised by the reviewer. Furthermore, we will add a note on the long-term treatment strategy to the discussion section in accordance with your suggestion. (Page 6, Line 198-200)

Reviewer 3 Report

The study was carried out with high scientific soudness.

A variety of detailed results of electrostimulation of FDL and FHL after BoNT injection for treatment CFD in patients with cerebral infarction and patients with cerebral hemorrhage are presented.

At least the results and classification appear to be of a very theoretical nature.

Author Response

Manuscript ID: toxins-1888326

Title: Effects of flexor digitorum longus muscle anatomical structure on the response to botulinum toxin treatment in patients with post-stroke claw foot deformity

Dear 

Thank you for your e-mail regarding the decision on the above manuscript. We were pleased to know of your positive evaluation of our manuscript and its potential acceptance for publication in Toxins, subject to adequate revision and response to the reviewer's comments. 

Based on your instructions, we logged into the Editorial Manager website and submitted the marked up file of the revised manuscript (file name: toxins-1888326-R1m), the clear file of the revised manuscript (file name: toxins-1888326-R1) and the file of the point-by-point response to the comments raised by the reviewers (file name: toxins-1888326-rev) in Microsoft Word format.

Appended to this letter is our detailed point-by-point response to the comments raised by the reviewers. We agreed with all the comments. We revised the text of the discussion section primarily according to the reviewer's suggestions, especially with regard to the clinical relevance. In addition, in order to make the manuscript is easier to read, we revised the English text a little, taking care not to change the main findings and discussion of the results.

We take this opportunity to express our gratitude to the reviewer for the constructive and useful remarks. The comments allowed us to identify areas in our manuscript that needed modification and clarification. We also thank you for allowing us to resubmit a revised copy of the manuscript.

I hope that the revised manuscript is now acceptable for publication in Toxins.   

Sincerely Yours,

Manuscript ID: toxins-1888326

Title: Effects of flexor digitorum longus muscle anatomical structure on the response to botulinum toxin treatment in patients with post-stroke claw foot deformity

Point-by-point response to the comments of Reviewer #3

We thank the reviewer for evaluating our manuscript. The following text describes our response to the comments made by the reviewer. All line numbers mentioned in each response to each comment refer to the small-size numbers that appear on the left margin of the text of the revised manuscript.

Comments to the authors:

The study was carried out with high scientific soudness.

A variety of detailed results of electrostimulation of FDL and FHL after BoNT injection for treatment CFD in patients with cerebral infarction and patients with cerebral hemorrhage are presented.

At least the results and classification appear to be of a very theoretical nature.

We thank the reviewer for the positive evaluation of our manuscript.

The manuscript was revised according to the comments raised by the reviewer. We look forward to your continued guidance in the future.
